# Developmental Profile in Children Aged 3–6 Years: Down Syndrome vs. Autism Spectrum Disorder

**DOI:** 10.3390/bs14050380

**Published:** 2024-04-30

**Authors:** Patricia López Resa, Esther Moraleda Sepúlveda

**Affiliations:** 1Departamento de Psicología, Facultad de Ciencias de la Salud, Universidad de Castilla La Mancha, 45600 Talavera de la Reina, Spain; 2Departamento de Psicología Experimental, Procesos Cognitivos y Logopedia, Facultad de Psicología y Logopedia, Universidad Complutense de Madrid, 28040 Madrid, Spain; esmora01@ucm.es

**Keywords:** early childhood, cognitive development, language, Down Syndrome, autism spectrum disorder

## Abstract

This research aims to compare the developmental profiles of children with autism spectrum disorder (ASD) and children with Down Syndrome (DS) between the ages of 3 and 6 years. The study examines whether these developmental disorders share common developmental milestones or exhibit distinctive characteristics. A total of 43 children, 23 with DS and 20 with ASD, participated in the study. Cognitive and language skills were assessed using standardized tools, including the Battelle Developmental Inventory, Reynell Developmental Language Scales III, and NEPSY-II battery. The results indicated that children with ASD outperformed children with DS in the areas of fine motor skills, gross motor skills, and communication. Additionally, children with ASD demonstrated higher scores in language comprehension and expressive language, compared to children with DS. Significant correlations were found between motor skills and communication abilities. Neuropsychological evaluations revealed significant differences between the two groups in various tasks, such as the comprehension of instructions, body part naming and identification, and recognition of emotions. These findings contribute to our understanding of the similarities and differences between ASD and DS, shedding light on the dissociation between cognition and language and its impact on adaptive functioning in these populations.

## 1. Introduction

Currently, there is a growing body of evidence highlighting the significance of early childhood in human development, particularly in the domains of cognitive and linguistic development [1]. Given that the acquisition of these skills follows a predictable and common sequence in typically developing (TD) children, we can assert that, in certain instances, there are discernible regular patterns in the milestones of development [2,3,4,5,6]. However, this does not necessarily hold true in neurodevelopmental disorders [7,8], where we observe a range of deviations from the typical sequence of milestone acquisition. This can manifest as a temporary delay in the acquisition sequence, as seen in Down Syndrome [9,10], or as atypicalities in the pattern of acquisition, as observed in autism spectrum disorder [11,12] 

Regarding Down Syndrome (hereafter referred to as DS), it is the most common identifiable genetic cause of intellectual disability [13,14]. It is a neurodevelopmental disorder that occurs in 1 in every 732 live births, although its prevalence has decreased over time, due to prenatal diagnosis [15]. Clinically, DS is characterized by its own phenotypic profile [16], variable intellectual disability [16], alterations in development and organic/systemic functioning [17], and impairments in the acquisition and development of communication and language [18]. However, despite a widespread consensus on the existence of developmental alterations in this population, not all domains appear to be equally affected [19]. In this regard, phonological memory, reasoning abilities, motor development, and prelinguistic elements acquisition are particularly compromised [20,21], leading to a cognitive profile that negatively impacts language [22] and adaptive functioning in the DS population [23]. Studies conducted to date have identified strengths in DS, such as socialization, motor skills, and activities of daily living [24,25], as well as weaknesses in communication [23].

On the other hand, autism spectrum disorder (ASD) is defined as a neurodevelopmental condition characterized by deficits in social communication, as well as the presence of restricted interests and repetitive behaviors [26]. It is a heterogeneous set of alterations that emerges in childhood and persists throughout the lifespan of the individual [27]. The worldwide prevalence is approximately 1%, being four times more common in males than females [28,29]. Regarding cognitive level, not all individuals with ASD present associated intellectual disability [30]. Furthermore, numerous studies have shown that most individuals with ASD, regardless of IQ, exhibit peak performance, for example, in block design tasks, and poorer performance in tasks involving verbal comprehension, revealing a discrepancy between verbal and non-verbal intelligence [31,32,33].

Regarding communication competence in ASD, a considerable number of researchers have focused their efforts on delineating the conceptions of “nonverbal” and “minimally verbal” children [34,35], thus demonstrating the existence of great heterogeneity in the overall development of children with ASD [36,37]. Furthermore, some authors report that between 30% and 40% of individuals with ASD do not acquire oral language as a form of communication [38], which negatively interferes with their social development and adaptive functioning [39]. Additionally, some studies have indicated that, while there is a significant correlation between linguistic and cognitive abilities, as well as their impact on adaptive functioning, higher levels of nonverbal intelligence are not always associated with linguistic level [40] or, consequently, better adaptive functioning [41]. On the other hand, Logan et al. [42] demonstrated that frequent delays in the acquisition and development of linguistic skills, such as word imitation, lack of protoconversations, vocabulary production, and responding to one’s own name, are the characteristics that most draw the attention of early age child interlocutors.

Therefore, the findings related to the childhood development of both populations suggest, to some extent at least, a dissociation between cognition and language that interferes with individuals’ adaptive functioning, which becomes particularly evident when comparing the performance of children with Down Syndrome (DS) and ASD [43]. For example, concerning cognition, a study conducted by Adamson et al. [44] found that children with ASD exhibited a deficit in establishing sustained joint attention, whereas, although the DS group could establish joint attention, difficulties arose in conveying desire or intention about a specific object. Regarding communication skills, a study by Sigman et al. [45] found that social interaction, communication, and joint attention skills improved as the chronological age increased, in groups composed of individuals with DS and ASD. Additionally, as the chronological age increased, children in both groups exhibited a greater number of initiations of play behaviors and a decrease in rejecting play requests. However, when the groups were matched for mental age, these differences disappeared, suggesting that cognitive capacity may have influenced these results.

Regarding the assessment of adaptive functioning profiles at early ages, studies such as that by Fidler and Nadel [46] have shown that when children with Down Syndrome (DS) are matched for mental age with other developmental disabilities, they obtain higher scores, demonstrating statistically significant differences compared to other disabilities. In this vein, the work of [47], after evaluating adaptive functioning in children under 36 months with ASD and other developmental disabilities, found lower socialization and communication skills in the group of children with ASD, as well as greater discrepancies between adaptive functioning and mental age.

On the other hand, it appears that individuals with DS may exhibit characteristics similar to those with ASD. For example, Kirchner and Walton [48] compared a group of 44 individuals with ASD, 78 with DS, and 46 with Williams Syndrome (WS), all aged between 6 and 18 years, who were assessed using the Social Communication Questionnaire. The results of the study showed that both individuals with DS and those diagnosed with WS obtained high scores in all symptomatic areas of ASD. Additionally, the DS group scored significantly lower on the subscales of unusual behaviors, social communication, and self-regulation, compared to individuals with ASD, while no statistically significant differences were found in the areas of language and atypical behaviors.

Therefore, the scientific literature has not yet clearly elucidated the similarities and differences between both developmental disorders. Given the limited understanding of the common characteristics of both disorders, the aim of the present research has been to compare the developmental profiles of the population with ASD and the population with DS, during the specific developmental stage of 3 to 6 years. This study aims to observe whether the profiles of both disorders follow common developmental milestones or whether DS and ASD exhibit distinctive characteristics from each other.

## 2. Method

### 2.1. Participants

A total of 43 children (29 girls and 14 boys) between the ages of 3 and 6 years, divided into two groups, constituted this study. The first group consisted of 23 children with Down Syndrome (14 girls and 9 boys), with a mean chronological age of 3.75 years (DS = 0.81). The second group consisted of 20 participants diagnosed with autism spectrum disorder Level 2 accompanied by associated intellectual disability (14 girls and 6 boys), with a mean age of 3.61 years (DS = 0.44).

Furthermore, since the participants’ intellectual capacity could explain differences in developmental profiles, the cognitive development level assessed using the Battelle Developmental Inventory was analyzed first. No significant differences were found between the two groups (*p* < 0.05). Additionally, we ensured that participants did not present comorbid diagnoses, by reviewing assessment and diagnostic reports provided by the parents.

### 2.2. Instruments

To conduct this study, cognitive and language skills were assessed using various standardized tools. Firstly, the Battelle Developmental Inventory [49] was used to assess the overall development of the children in the study. The inventory consists of 341 items, grouped into the domains of Personal/Social, Adaptive, Motor (gross and fine motor skills), Communication (Receptive and Expressive), and Cognitive. Its purpose is to gather data on strengths and weaknesses in children’s developmental areas (from 0 to 95 months of age), facilitating the creation of personalized intervention programs. Results are expressed in terms of Developmental Age Equivalents for each area, although they can also be converted into Developmental Quotients, to compare results obtained by individuals of different ages. Furthermore, it demonstrates reliability indices for various scores and age ranges, ranging from 0.88 to 0.99.

Secondly, the Reynell Developmental Language Scales III [50] were employed to evaluate language. These scales consist of 62 items divided into the following two scales: comprehension and expression. The comprehension scale assesses various aspects of language, such as preverbal and affective concepts, object naming, question interpretation, and verbal reasoning. On the other hand, the expressive language scale focuses on language structure, vocabulary, and content. This test is suitable for individuals between 18 months and 7 years old and is useful for evaluating both comprehension and expressive language development, facilitating the planning of necessary interventions. Additionally, it provides reliability measures across different score ranges and age brackets, ranging from 0.44 to 0.96.

Lastly, the NEPSY-II battery [51] was used, which allows for the assessment of neuropsychological functioning in children aged 3 to 16 years through 36 tests that are grouped into the following 6 domains: (1) Attention and Executive Function, (2) Language, (3) Memory and Learning, (4) Social Perception, (5) Visuospatial Processing, and (6) Sensorimotor. The NEPSY-II provides internal consistency coefficients and reliability scores for test scores across different age groups, with these coefficients ranging from 0.60 to 0.99. Additionally, inter-rater reliability studies yielded highly positive results, with agreement percentages ranging from 76% to 99%, depending on the subtest.

### 2.3. Procedure

To initiate the research, contact was initially made with the Down Syndrome Federation of Castilla-La Mancha, as well as various healthcare centers and private clinics in Castilla-La Mancha, Community of Madrid, and Castilla y León, to inform them of the project and to determine their interest in participating in the present study. All contacted centers expressed their willingness to participate in the study. After confirming the subjects for the sample, the participants were sent the informed consent document. Once signed, the work schedule was established, agreeing on the dates of the assessments. All participants were evaluated in four individual sessions of 45 min each. Questionnaires on Quality of Life and Adaptive Functioning were completed by the parents during the first session. The research was conducted with the explicit consent of the parents. All participants had either a diagnosis of Down Syndrome made by a doctor, or a diagnosis of autism spectrum disorder previously made by a psychologist. The informed consent and the project were approved by the Ethics Committee in Social Research of the integrated area of Talavera de la Reina, with registration number CAU-683200-X6H7.

### 2.4. Data Analysis

IBM SPSS 24.0 Statistics Multilingual software was used as the tool for data analysis. To determine the degree of similarity between the sample distribution and the population distribution and, thus, to test the normality of the sample, a goodness-of-fit test was performed using the Kolmogorov–Smirnov test, obtaining parametric variables (>0.5) and non-parametric variables (<0.5). Additionally, to determine if there were significant differences between the different variables among the two groups within the age range of 3–6 years, Levene’s test for equality of variances and the *t*-test for equality of means were employed for parametric variables, as they are quantitative variables with a normal distribution. For non-parametric variables, which are quantitative variables without a normal distribution, the Kruskal–Wallis analysis of variance test was used. Spearman’s rho test was utilized to analyze the correlation between different areas for non-parametric variables.

## 3. Results

The most relevant results regarding the groups of children with DS and ASD are presented below.

Regarding the Battelle Developmental Inventory, as shown in Table 1, children with ASD obtained higher scores in the Adaptive, Fine Motor, Total Motor (Fine Motor + Gross Motor), Communication, Personal, and Cognitive domains, only scoring lower than the group of children with DS in the Gross Motor area. However, significant differences were found in favor of the group of children with ASD in the Gross Motor, Total Motor, and Communication domains (*p* < 0.05).

On the other hand, data obtained from the Reynell-III Scale (Table 2) illustrated how children with ASD outperformed the group of children with DS, in the areas of Total Score, Age Equivalent in years, Comprehension, and Expression. However, significant differences were found in the expressive domain (*p* < 0.05).

The results obtained from the correlation analysis between the different assessed areas of both tests (Table 3) revealed a significant correlation between the motor domain and the communication domain, as well as between the motor domain and the expressive level (*p* < 0.01).

Regarding the results of the neuropsychological evaluation conducted using the NEPSY-II tool, Table 4 presents the descriptive statistics.

We can observe a profile with statistically significant differences between the groups of children with DS and with ASD, respectively. In this regard, significant differences were found in the tasks of comprehension of instructions (F(1, 38) = 3.265; *p* < 0.01; h2 = 0.079), naming body parts (F(1, 38) = 48.173; *p* < 0.001; h2 = 0.559), identifying body parts (F(1, 38) = 40.741; *p* < 0.001; h2 = 0.517), recognition of emotions (F(1, 38) = 6.840; *p* < 0.01; h2 = 0.153), copying designs (F(1, 38) = 20.987; *p* < 0.01; h2 = 0.356), and imitation of hand configurations (F(1, 38) = 25.831; *p* < 0.01; h2 = 0.383), where the group of children with ASD obtained higher scores than the DS group, as well as in the tasks of pseudoword repetition (F(1, 38) = 28.148; *p* < 0.01; h2 = 0.426) and oromotor sequences (F(1, 38) = 17.255; *p* < 0.01; h2 = 0.404), where the DS group obtained higher scores. No significant differences were found in the remaining variables, except for theory of mind, which could not be considered because the scores were 0 in all cases.

## 4. Discussion

The main objective of this study was to investigate the differences that could be found in the developmental profiles between children diagnosed with ASD and DS in the 3–6 years age range, given the scarcity of research conducted in this specific age group. The results, in general, have revealed significant differences in the developmental profiles of both populations.

Specifically, the data obtained from the Batelle Developmental Inventory have demonstrated better motor and communicative skills in the group of children with ASD compared to the group with DS, in contrast to other studies [43]. Regarding the motor component, some studies have shown better motor skill performance in children with ASD, compared to children with DS [52,53,54], which is in line with our data. However, both populations seem to present motor difficulties that arise because of sensory processing difficulties [21,55], such as perceptual–motor coupling [56,57]. For example, authors like Simon et al. [58] found difficulties in performing tasks such as grasping, throwing, and catching, which seemed to affect the establishment of essential interaction formats in language acquisition in a group of young children with DS [59,60]. On the other hand, Tomasello [61] reported the existence of perceptual biases in infants that guide attention towards people and objects, which, along with motor development enabling exploration, lead to the emergence of secondary intersubjectivity towards the end of the first year of life. Therefore, we find some studies suggesting certain implications of a child’s motor development on their language acquisition [62,63,64,65,66], in line with the correlation between the motor component and the communicative–linguistic component found in this study.

Regarding language development specifically, the Reynell Language Development Scales III [50] and the NEPSY-II Battery [51] showed better comprehension of instructions and naming and identifying body parts skills in the group of children with ASD. On the other hand, our group of children with DS obtained higher scores in oromotor sequence tasks and pseudoword repetition. However, nowadays, we can assert that there is some consensus in the scientific literature that ASD does not imply a primary deficit at the phonological level [67]. In this regard, several studies have shown the existence of phonological and articulatory difficulties [68,69]. All these data seem to contradict our results because, while it would be expected based on the literature that children with ASD would obtain better scores in articulatory tasks and tasks involving the acoustic–phonological conversion mechanism, our data indicate that it is the children with DS who obtain higher scores. This can be explained, to some extent, by the difficulties in the acquisition and development of imitation skills characteristic of ASD [70,71]. As such, a study conducted by Stone et al. [72] found that, during the early years of life, motor imitation skills positively correlated with expressive language abilities in children with ASD. In this respect, the data obtained in this research have also reported a greater naming ability in the group of children with ASD compared to the group of children with DS. These data can be interpreted in line with studies reporting that, while the lexical–semantic component seems to be a strong point in children with DS [10,73], it is altered compared to typical development [74]. In the same direction, two meta-analyses, focused on the lexical–semantic skills of children with ASD, have highlighted a delay in vocabulary acquisition compared to typical development [75,76]. All these data suggest that, while both populations present linguistic difficulties compared to typical development, children with ASD seem to have fewer difficulties in decontextualized tasks that do not require the use of environmental information, as required in imitation and repetition tasks [77,78]. 

It is particularly striking that the group of children with ASD presents higher expressive communicative competence, especially in the 3–6-year-old stage, considering that communication difficulties are one of the distinctive features of this disorder [79]. These results contrast with those of Dardani et al. [80], who reported that children with DS have a higher level of communicative and linguistic skills compared to children with ASD. However, our data can be interpreted in line with studies that have shown a greater impairment in receptive skills compared to expressive skills, in preschool-aged children diagnosed with ASD [81,82]. In this respect, the meta-analysis conducted by Kwok et al. [77] revealed that, although there seems to be an expressive advantage over receptive skills, it is not a common profile within the ASD population, so this discrepancy in language should not be considered a marker of ASD [83,84]. All these findings could explain, at least to some extent, the significant expressive differences between the two groups, as there are several studies that have documented a significant delay in language acquisition in people with DS [85,86]. For example, a study conducted by Buckley and Oliver [87] found that the mean age for the consolidation of two-word combinations in children with DS occurs at around 37 months, presenting a delay of 18 months compared to normotypic development. On the other hand, studies on ASD show even greater discrepancies, specifically due to the variability in the symptomatology of ASD. In this respect, Rose et al. [88] found, in a sample of 246 children diagnosed with ASD, that approximately 36% were not using two-word phrases at 40 months, compared to 64% who were.

Some studies conducted between DS and ASD point to uneven development in different areas of language. For example, Roberts et al. [10] compared language and communication development in children with DS and ASD aged between 3-and-a-half years and 16 years, with similar cognitive abilities. The authors found that children with DS showed better performance in language comprehension and pragmatics, while children with ASD showed better performance in expressive language. On the other hand, a study by Laws, Bishop, and Adams [89] found that children with DS performed better in language comprehension and sentence production, while children with ASD performed better in passive vocabulary and receptive language in general. Similarly, de Falco et al. [90] found that, while children with DS had better pragmatic and linguistic comprehension skills, the sentence production and expressive language of children with ASD showed a higher performance, in line with the findings of this research.

Therefore, the current findings suggest an influence of the severity of prototypical symptomatology in ASD on the relative receptive and expressive language abilities [82,83,91]. In this regard, our esults seem to indicate that, when equating children with DS and children with ASD with intellectual disability (ID) based on cognitive development, children with ASD in the 3–6-year-old stage appear to have a better expressive language capacity. These new findings, taken into consideration along with the information found in the scientific literature on the developmental trajectories of both populations, emphasize the need to open lines of research that focus their efforts on comparing the clinical phenotype of ASD and genetic syndromes such as DS in order to obtain high-quality information that enables the creation of assessment protocols tailored to the needs of each population, as well as effective intervention programs.

However, it is important to highlight that the present study is not without limitations. Firstly, it should be noted that participants in the ASD group were recruited based on the researchers’ review of diagnostic reports. Therefore, the lack of confirmation of the diagnosis could be considered as a confounding variable that may interfere with the results. Secondly, the atypical gender distribution within the ASD group sample is noteworthy. In this regard, while the scientific literature reports a significantly higher prevalence of ASD in males than females, most participants in our study were female. This could potentially complicate the interpretation of the results.

## Figures and Tables

**Table 1 behavsci-14-00380-t001:** Descriptive statistics obtained in the Battelle Scale.

Domain	Means for DS Group	Means for ASD Group	*p*-Value
Adaptive domain	68.74 (16.12)	74.42 (7.44)	0.251
Gross motor domain	44.39 (33.47)	33.84 (21.89)	0.005 *
Fine motor domain	27.48 (12.77)	36.53 (3.61)	0.113
Total motor domain	52.22 (38.79)	76.47 (22.94)	0.011 *
Communication domain	46.91 (17.32)	56.05 (7.92)	0.006 *
Personal domain	110.13 (27.53)	117.47 (9.79)	0.276
Cognitive domain	28.74 (21.29)	41.05 (16.70)	0.197

Note: Standard deviations in parentheses. * *p* > 0.005.

**Table 2 behavsci-14-00380-t002:** Descriptive statistics obtained in the Reynell-III Scale.

Domain	Means for DS Group	Means for ASD Group	*p*-Value
Total score	321.04 (107.12)	372.05 (33.47)	0.168
Age Equivalent in years	2.57 (1.02)	2.97 (0.33)	0.187
Comprehension	23.74 (11.81)	34.74 (8.14)	0.078
Expression	6.48 (3.32)	6.89 (2.82)	0.02 *

Note: Standard deviations in parentheses. * *p* < 0.05.

**Table 3 behavsci-14-00380-t003:** Correlations between the Battelle Developmental Inventory and the Reynell-III Scale.

	Adaptative Domain (BDI)	Gross Motor Domain (BDI)	Fine Motor Domain (BDI)	Total Motor Domain (BDI)	Communication Domain (BDI)	Personal Domain (BDI)	Cognitive Domain (BDI)	Comprehension (RS)	Expression (RS)
**DS group**									
Adaptative domain (BDI)									
Gross motor domain (BDI)					*				
Fine motor domain (BDI)									
Total motor domain (BDI)					**				**
Communication domain (BDI)		*		**					
Personal domain (BDI)									
Cognitive domain (BDI)									
Comprehension (RS)									
Expression (RS)		*		**					
**ASD group**									
Adaptative domain (BDI)									
Gross motor domain (BDI)					*				
Fine motor domain (BDI)									
Total motor domain (BDI)					**				**
Communication domain (BDI)		*		**					
Personal domain (BDI)									
Cognitive domain (BDI)									
Comprehension (RS)									
Expression (RS)		*		**					

* *p* < 0.05; ** *p* < 0.01.

**Table 4 behavsci-14-00380-t004:** Descriptive statistics obtained from the NEPSY-II assessment.

	Domain	Means for DS Group	Means for ASD Group	*p*-Value
Attention and Executive Function	Statue	8.85 (0.36)	8.35 (1.18)	0.213
	Inhibition	3.00 (0.75)	4.90 (1.74)	0.325
Language	Comprehension of Instructions	3.00 (0.72)	4.90 (5.00)	<0.01 *
	Naming Body Parts	2.65 (2.76)	7.50 (1.47)	<0.001 *
	Identifying Body Parts	1.85 (2.03)	5.00 (0.85)	<0.001 *
	Semantic Verbal Fluency	2.90 (0.30)	2.85 (0.67)	0.065
	Phonological Verbal Fluency	1.05 (0.22)	1.25 (0.55)	0.140
	Phonological Processing	1.35 (1.60)	1.20 (1.06)	0.123
	Pseudoword Repetition	22.25 (2.63)	7.10 (3.12)	0.423
	Oromotor Sequences	4.55 (1.63)	2.65 (1.22)	0.185
	Naming Speed	1.50 (0.88)	1.70 (0.65)	0.303
Memory and learning	Face Memory	4.10 (1.16)	6.90 (1.86)	0.215
	Design Memory	6.50 (2.48)	8.70 (2.25)	0.314
	Narrative Memory	4.05 (0.22)	3.55 (0.51)	0.145
	Word List Interference	2.40 (0.82)	3.50 (0.70)	0.185
	Phrase Repetition	1.75 (1.16)	2.50 (1.27)	0.723
Social perception	Recognition of Emotions	0.20 (0.61)	1.10 (1.41)	<0.01 *
	Theory of Mind	0 (0)	0 (0)	-
Visuospatial Processing	Block Construction	2.45 (1.27)	2.10 (0.78)	0.303
	Design Copying	3.60 (0.99)	4.90 (0.78)	<0.01 *
Sensorimotor	Imitation of Hand Configurations	4.00 (0)	5.10 (0.96)	<0.01 *
	Visuomotor Precision	2.80 (0.89)	3.20 (1.70)	0.359

Note: Standard deviations in parentheses. * *p* < 0.05.

## Data Availability

Data are contained within the article.

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
