# Peer review of "Developmental Profile in Children Aged 3–6 Years: Down Syndrome vs. Autism Spectrum Disorder"

_behavsci, 2024, doi:10.3390/bs14050380_

Round 1

Reviewer 1 Report

Comments and Suggestions for Authors

While this comparison between young children with autism and those with Down is interesting as they are matched on cognition it only adds limited information to the literature. I am also concerned that there seems to have been no attempt to ensure that the children with DS did not also have ASD as it is common in 18% or more of these children. Some of the references do not seem to refer to point being made e.g. Van Braeckal et al 2010 p 2. Annaz et al 2009. I recommend a careful re read of the paper and the checking of accuracy of references. My main concern is the presentation of the findings. The Tables should specify what the numbers given represent as measures. In Table 1 the DS gross motor score appears higher than for ASD yet is reported as favouring ASD. The Reynell data would be better in a Table and all statistical test data better presented. The data on correlations between motor and communication scores should be in a Table - and if confirmed, are relevant for planning early interventions. On the NEPSY-II I find the better oromotor scores and pseudoword rep score advantaging DS very surprising given the speech-motor challenges that most children with DS have in this age range - maybe check on this.

In the discussion - again I have some concerns that papers referenced may not actually support the points being made and should be checked. I note SD is used frequently when in English it is DS. 

Comments on the Quality of English Language

Is largely good but may be will benefit from a check 

Author Response

Dear reviewer, thank you for all your contributions. Attached below is a Word document containing responses to your comments.

Reviewer 2 Report

Comments and Suggestions for Authors

The way the data collected in this research are analyzed and discussed is highly questionable. The paper needs an almost complete overhaul to be acceptable for publication. More in detail:

1.    The psychometric tests  used recover partially each other. You need to compute the correlations between the subtests and check whether they are sufficiently in agreement. If not, this may be a problem for your interpretation of the data.

2.    The category ASD is clinically useful. However, in a comparative work it may be somewhat misleading. ASD subsumes also Asperger syndrome. It is known that many children with AS have a good formal language and develop it sometimes even more rapidly than typical children. Did your ASD sample contain AS children. In any case, you need to specify better your ASD sample as to the composing subgroups of children. If indeed your sample contained children with AS, your comparative results could be explained, at least partially more economically, as confirming a well-known indication, i.e., young ASD-AS children have better language and communication skills than young DS children of about the same cognitive levels

3.    Psychometric tests are not sufficient to make a judgement on the functional language abilities of the language users. To be sure, you need collecting samples of spontaneous speech and language and analyze them in detail (mean length of utterances, morphosyntax, relational semantics, lexicon, pragmatic skills, etc.).

4.    More on the data in the paper, why bother reporting lengthy indications on nonsignificant results. By definition, they are not guaranteed beyond the samples in the populations of subjects.

5.    As to the statistical analysis, what is your logical basis for mixing parametric and nonparametric tools? As you know, parametric tests demand normality of distribution and homoscedasticity, whereas nonparametric tools dispense with a strict respect of these conditions. By the way, you mention “parametric and nonparametric variables”. Statistics may be parametric or nonparametric, variables are just variables.

Author Response

(The authors gave the same response as above.)

Round 2

Reviewer 1 Report

Comments and Suggestions for Authors

The authors seem to have satisfactorily addressed all the points that I raised in my review - and corrected errors 

Author Response

Dear Reviewer,

Thank you once again for your time and feedback.